# Identification of natural Zika virus peptides presented on the surface of paediatric brain tumour cells by HLA class I

**Matt Sherwood[1], Ben Nicholas[1], Alistair Bailey**[1,2]**, Thiago Giove Mitsugi[3], Carolini Kaid[3], Oswaldo K. Okamoto[3]\*, Paul Skipp**[1]**, Rob M. Ewing**[1]

**1** Centre for Proteomic Research, Biological Sciences and Institute for Life Sciences, Building 85, University of Southampton, United Kingdom, **2** Centre for Cancer Immunology and Institute for Life Sciences, Faculty of Medicine, University of Southampton, United Kingdom, **3** Centro de Estudos do Genoma Humano e Células-Tronco, Departamento de Genética e Biologia Evolutiva, Instituto de Biociências, Universidade de São Paulo, Cidade Universitária, São Paulo, S.P., Brazil

☯ Equal contributions
\* rob.ewing@soton.ac.uk

**Data availability statement:** The mass spectrometry proteomics data have been deposited to the ProteomeXchange Consortium via the PRIDE partner repository with the dataset identifier PXD037627 and 10.6019/PXD037627.

## Abstract

Despite decades of research, survival from brain cancer has scarcely improved and is drastically lower than that of other cancers. Novel therapies, such as immunotherapy, hold great promise for treating brain tumours and are desperately needed. Zika virus (ZIKV) infects and kills aggressive cancer cells with stem-like properties (CSCs) from both paediatric and adult brain tumours. Whilst T cell recruitment into ZIKV-infected brain tumours is becoming well documented, the specific mechanisms through which they are activated are poorly understood. We address this by employing a combined global proteome and immunopeptidome mass spectrometry approach to describe, for the first time, human leukocyte antigen (HLA) presentation of ZIKV peptides on the surface of infected brain tumour cells. We first show that antigen processing and presentation by HLA class I (HLA-I) is the top enriched immune response pathway in the global proteome of aggressive paediatric USP7-ATRT brain tumour cells following ZIKV infection. We identify USP7-ATRT cells as a desirable immunopeptidome model as they express the globally common HLA-A allotype (A\*02:01). We predict the majority of our 19 identified ZIKV peptides to strongly bind and be presented by HLA-A\*02:01. We observe a trend between immunopeptide presentation and cellular ZIKV protein abundance, with nearly half of the peptides arising from the most abundant viral protein; non-structural protein 3 (NS3). We show the ZIKV NS3 helicase domain to be a particularly rich source of peptides. Finally, we verify that the 19 ZIKV peptides identified here are not predicted to mimic peptides of the human proteome. The ZIKV peptides we identify here are novel targets for immunotherapy, and our findings provide potential insight into the efficacious cytotoxic T cell response that oncolytic ZIKV virotherapy can induce against brain tumours.

**Funding:** This project was funded in part by the Medical Research Council (MRC) award number MR/S01411X/1 to R.M.E. and additional funding provided by Neuroblastoma UK, The Little Princess Trust, the Rosetrees Trust and Wessex Medical Trust (all to R.M.E.). The funders had no role in study design, data collection and analysis, decision to publish, or preparation of the manuscript.

**Competing interests:** I have read the journal's policy and the authors have the following competing interests: Oswaldo K. Okamoto is an Advisor at Vyro biotherapeutic Carolini Kaid is an employee at Vyro biotherapeutic.

## Introduction

Central nervous system (CNS) tumours account for approximately one-fifth of all childhood cancer cases and are disproportionately the largest cause of cancer-related mortality in children [1,2]. These tumours exhibit high lethality, and the aggressive nature of standard-of-care therapy often leaves survivors with severe sequelae that significantly affect their quality of life. CNS tumours often present with a suppressive tumour immune microenvironment (TIME) through a combination of intrinsic (reduced antigen presentation, immune checkpoint blockade and immunosuppressive cytokine secretion) and extrinsic (immunosuppressive immune cell recruitment) factors [3]. There is significant interest in developing immunotherapeutic strategies to circumvent this suppression by activating a patient's immune system against their tumour [4].

Oncolytic virotherapy, a specific class of immunotherapy, exploits viruses that preferentially infect and destroy tumour cells with minimal pathology against non-cancerous cells and tissues. The second pillar of oncolytic virotherapy is the mounting of anti-tumoral immune responses following immunogenic cell death (ICD) of cancer cells. Tumour antigens, viral antigens, cytokines, pathogen-associated molecular patterns (PAMPs), and damage-associated molecular patterns (DAMPs) are released into the tumour microenvironment (TME) during OV-induced ICD, leading to inflammation [5]. Initially, these factors recruit and stimulate innate immune cells such as monocytes, macrophages, dendritic cells (DCs), neutrophils, and natural killer (NK) cells. Professional antigen-presenting cells (APCs) such as macrophages and DCs bridge innate and adaptive immunity by processing and presenting tumour and viral antigens on their cell surface. Adaptive immune system cells such as T cells migrate along chemokine and molecular gradients to locate, infiltrate and drive an adaptive immune response against the infected tumour. Thus, the efficacy of OV therapy arises from various components, including oncolysis, the innate immune response and the adaptive immune response. This gives OVs a unique advantage in targeting highly heterogeneous and immunosuppressive cancers, such as CNS tumours.

Oncolytic virotherapy clinical studies have generally reported low toxicity and minimal adverse effects in patients, and there are over 200 clinical trials underway to treat aggressive forms of cancer using OVs [6–8]. Recently, the oncolytic herpes virus G47Δ was approved in Japan for glioblastoma treatment, the first oncolytic virotherapy against any nervous system tumour in the clinic [9]. As OVs can remodel the immunosuppressive TIME, there is significant interest in employing them as adjuvants to other immunotherapies, including monoclonal antibodies, CAR-T cells, cancer vaccines, checkpoint inhibitors and small molecule inhibitors [10,11].

ZIKV is neuropathogenic and causes congenital ZIKV syndrome (CZS) in 5–14% of babies born to women who contract ZIKV during pregnancy and pass the virus to the fetus via transplacental transmission [12]. ZIKV infects and diminishes the pool of fetal neural stem and progenitor cells (NPCs) through induction of differentiation or cell death, subsequently leading to the underdevelopment of the fetal brain [13–16]. In contrast, postnatal ZIKV infection in children is mild, and only 1:5 people are symptomatic. ZIKV infection is generally self-limiting as symptoms resolve within a week

or less, and the majority of symptomatic children primarily present with flu-like symptoms [12,17]. This mild infection in children and ZIKV's neurotropism highlights the virus as a promising candidate for paediatric CNS tumour therapy.

Since 2017, members of our research team and others have demonstrated that ZIKV infects and induces oncolysis of paediatric brain tumour cells *in vitro* and *in vivo*, and mounts an immune response against spontaneous brain tumours in canines [18–20]. ZIKV infection stimulates the infiltration of multiple immune cell types into CNS tumours, including CD8+ and CD4+T cells, which contribute to ZIKV-induced tumour clearance [21–23]. This branch of the adaptive immune response is brought about by viral peptide presentation by the HLA class I and II on the cell surface to T cell receptors (TCRs) on CD8+ and CD4+T cells, respectively. HLA-presented peptide recognition and binding leads to TCR signalling, and co-stimulatory and co-inhibitory molecules govern the outcome of this signalling with regards to T cell function and fate [24]. Whilst the recruitment of T cells into ZIKV-infected CNS tumours is becoming well documented, their cognate HLA-presented ZIKV peptides remain unknown.

Previously, we demonstrated a aggressive paediatric atypical teratoid rhabdoid tumour (ATRT) cell line, USP7-ATRT, to have CSC properties and to be highly susceptible to ZIKV infection and oncolysis [19]. In the present study, we show that ZIKV infection enriches major histocompatibility complex (MHC) class I antigen processing and presentation at the proteome level in these paediatric brain tumour cells. To investigate this response further, we perform HLA typing to show that USP7-ATRT cells express all three classical HLA-I alleles and express the globally common HLA-A allotype (A*02:01) [25]. Performing immunopeptidome profiling, we identify a specific list of 19 ZIKV peptides from infected USP7-ATRT cells, predicted to be processed and presented by HLA-I molecules. The mass spectrometry proteomics and analysis used here are similar to our previous influenza work [26]. To our knowledge, we document ZIKV epitopes presented by human CNS tumour cells for the first time. We provide new ZIKV epitopes as novel targets for immunotherapy, and their identification should lead to future work that facilitates understanding of how the immune response can be coupled with the ZIKV oncolytic response.

## Results

### ZIKV infection enriches the HLA Class I presentation pathway in brain tumour cells

We first investigated whether ZIKV infection moderates innate or adaptive immune responses within the aggressive paediatric brain tumour cell line USP7-ATRT. Performing global proteome analysis of 24-hour ZIKV-infected USP7-ATRT cells and plotting the top and bottom 50 ranked proteins identified the ZIKV polyprotein as the second most highly ranked protein (Fig 1A). We thenperformed enrichment analysis using the set of 329 proteins that were found to be elevated ($p < 0.05$) at 24-hour following ZIKV infection vs control in this global proteome analysis. Using the Enrichr tool, we identified several pathway repositories in which immune-system related categories were highly significant: (i) NCATS BioPlanet - 'Immune System' and 'Antigen Presentation' categories were ranked 2nd and 7th (with adjusted p-values of 2.4 x $10^{-5}$ and 1.3 x $10^{-4}$) and (ii) Panther DB in which 'T-cell activation' was the 2nd most significant category ($p = 0.0071$) [27,28] (S1 File). To investigate in more detail, we performed Gene Set Enrichment Analysis (GSEA) with an innate and adaptive immune response specific Reactome database and observed the adaptive immune system "Class I MHC mediated antigen processing & presentation" term to be the most highly enriched pathway following ZIKV infection (Fig 1B and 1C). Supporting this, 20% of the 30 most highly ranked proteins from the ZIKV-infected USP7-ATRT samples (HUWE1, LTN1, PSMB5, PSMD1, RNF213 and SEC61G) are involved in antigen processing and presentation by MHC Class I (Fig 1A, S1 Table). To determine if this response is also observed for ZIKV-infected adult brain tumour cells, we repeated our GSEA analysis on a publicly available RNA-Seq dataset from eight glioblastoma patients comparing ZIKV-mCherry-positive versus -negative primary cell populations (Fig 1D). "Class I MHC mediated antigen processing & presentation" and "TCR Signalling" were the most highly enriched immune response terms in the ZIKV-mCherry positive glioblastoma cells (Fig 1D). To conclude, ZIKV infection enriches signatures indicative of antigen processing and presentation by MHC Class I in paediatric and adult brain tumour cells at the proteome and transcriptome levels, respectively. We sought to investigate this pathway further.

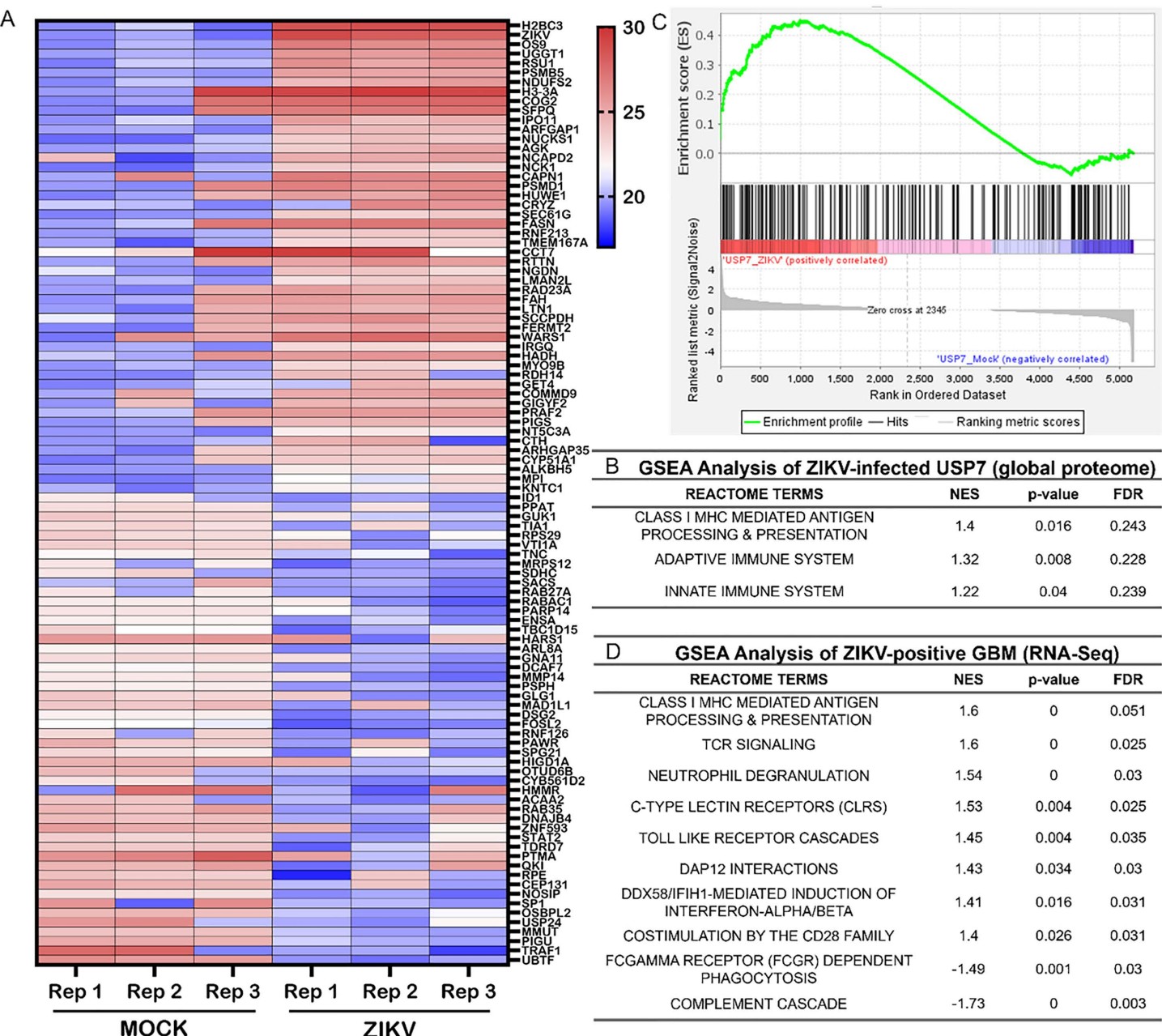

**Fig 1. ZIKV infection enriches the HLA Class I pathway in brain tumour cells. (A)** Heatmap plotting Log2(LFQ intensity) values of the top and bottom 50 ranked proteins in the ZIKV-infected USP7-ATRT cell global proteome. GSEA analysis to assess enrichment of innate and adaptive immune system Reactome pathways in **(B)** the ZIKV-infected USP7-ATRT cell global proteome (N = 3) and **(D)** the ZIKV-mCherry positive primary glioblastoma (GBM) cell transcriptome (N = 8). Normalised enrichment score (NES) denotes the degree to which the enrichment increased (+) or decreased (-). GSEA analysis significance is defined by p ≤ 0.05 and FDR ≤ 0.25. **(C)** GSEA enrichment plot of the significantly enriched "Class I MHC mediated antigen processing & presentation" term from the ZIKV-infected USP7-ATRT cell global proteome. The plot shows (i) the running enrichment score for the gene set (top), (ii) where the genes of the gene set appear in the ranked gene list (middle), and (iii) the change in the genes following infection (bottom). Abbreviations, Zika virus (ZIKV), USP7-ATRT (USP7), human leukocyte antigen (HLA), major histocompatibility complex (MHC), Gene Set Enrichment Analysis (GSEA), enrichment score (ES), normalised enrichment score (NES), false discovery rate (FDR), label-free quantitation (LFQ), glioblastoma (GBM).

## USP7-ATRT brain tumour cells predominately present HLA-A*02:01 peptides

Prior to investigating whether infection led to ZIKV-derived HLA immunopeptide presentation by USP7-ATRT cells, we first sought to understand what HLA allotypes USP7-ATRT cells express and what peptides these HLA molecules present. HLA typing of USP7-ATRT cell bulk RNA-Seq identifies that they express all three classical HLA-I alleles (Table 1). This analysis indicates that USP7-ATRT cells are homozygous for all three classical HLA-I alleles, but this may be a byproduct of HLA heterozygosity loss during transformation. USP7-ATRT cells express the globally most common HLA-A allotype (HLA-A*02:01), which is expressed by nearly 40% of the human population (Table 1). Additionally, USP7-ATRT expresses HLA-I molecules HLA-B*44:02 and HLA-C*05:01, and the HLA-II molecule HLA-DRB1*15:01 (Table 1). We demonstrate USP7-ATRT cells as a desirable immunopeptidome model due to their high HLA-A population coverage [29]. Performing immunopeptidomics, we assessed the class I and II HLA immunopeptidomes of Mock and ZIKV-infected USP7-ATRT cells (Table 2). Supporting our observations at the global proteome level, we predominantly observe peptides with lengths consistent with the nine amino acid (aa) preference for presentation by HLA-I rather than the longer peptides presented by HLA-II (Fig 2A). As expected for non-professional antigen-presenting cells, only modest numbers of HLA-II immunopeptides were recovered (Table 2). ZIKV infection did not affect the length distribution of the peptides presented (Fig 2A). Unbiased cluster analysis of all the distinct observed 9-mer peptides from ZIKV-infected USP7-ATRT cells identified 74%, 14%, and 12% of the 9-mer peptides to be presented by HLA-A*02:01, HLA-B*44:02 and HLA-C*05:01, respectively (Fig 2B). These are consistent with the USP7-ATRT HLA-I allotypes identified by HLA typing (Table 1). Peptide length distributions from ZIKV-infected USP7-ATRT cells show a dominance of 9-mer peptides, with differing minor populations of 8, 10 or 11-mer peptides, across the three classical HLA-I molecules (Fig 2C). To conclude, USP7-ATRT cells predominantly present 9-mer peptides by the three classical HLA-I molecules, with nearly three-quarters predicted to be presented by the globally most common HLA-A allotype HLA-A*02:01.

## USP7-ATRT brain tumour cells present ZIKV HLA-I immunopeptides, and presentation aligns with protein abundance in the HLA-I pathway

Next, we investigated whether infection led to ZIKV-derived HLA immunopeptide presentation and identified 19 HLA-I peptides derived from six of the ten ZIKV proteins (Table 3). Consistent with the low HLA-II expression by non-professional

**Table 1. USP7-ATRT cell HLA allotypes and their expression across the human population.**

| Locus | HLA allotype |
|---|---|
| HLA-A | A*02:01 |
| HLA-B | B*44:02 |
| HLA-C | C*05:01 |
| HLA-DRB1 | DRB1*15:01 |

HLA allotypes determined from RNA Sequencing of USP7-ATRT cells. Abbreviations, Zika virus (ZIKV), human leukocyte antigen (HLA).

**Table 2. Number of USP7-ATRT cell-presented ZIKV immunopeptides.**

| Condition | HLA class | Number of peptides |
|---|---|---|
| USP7-ATRT Mock | I | 3,866 |
| | II | 240 |
| USP7-ATRT ZIKV | I | 3,658 |
| | II | 222 |

Abbreviations, Zika virus (ZIKV).

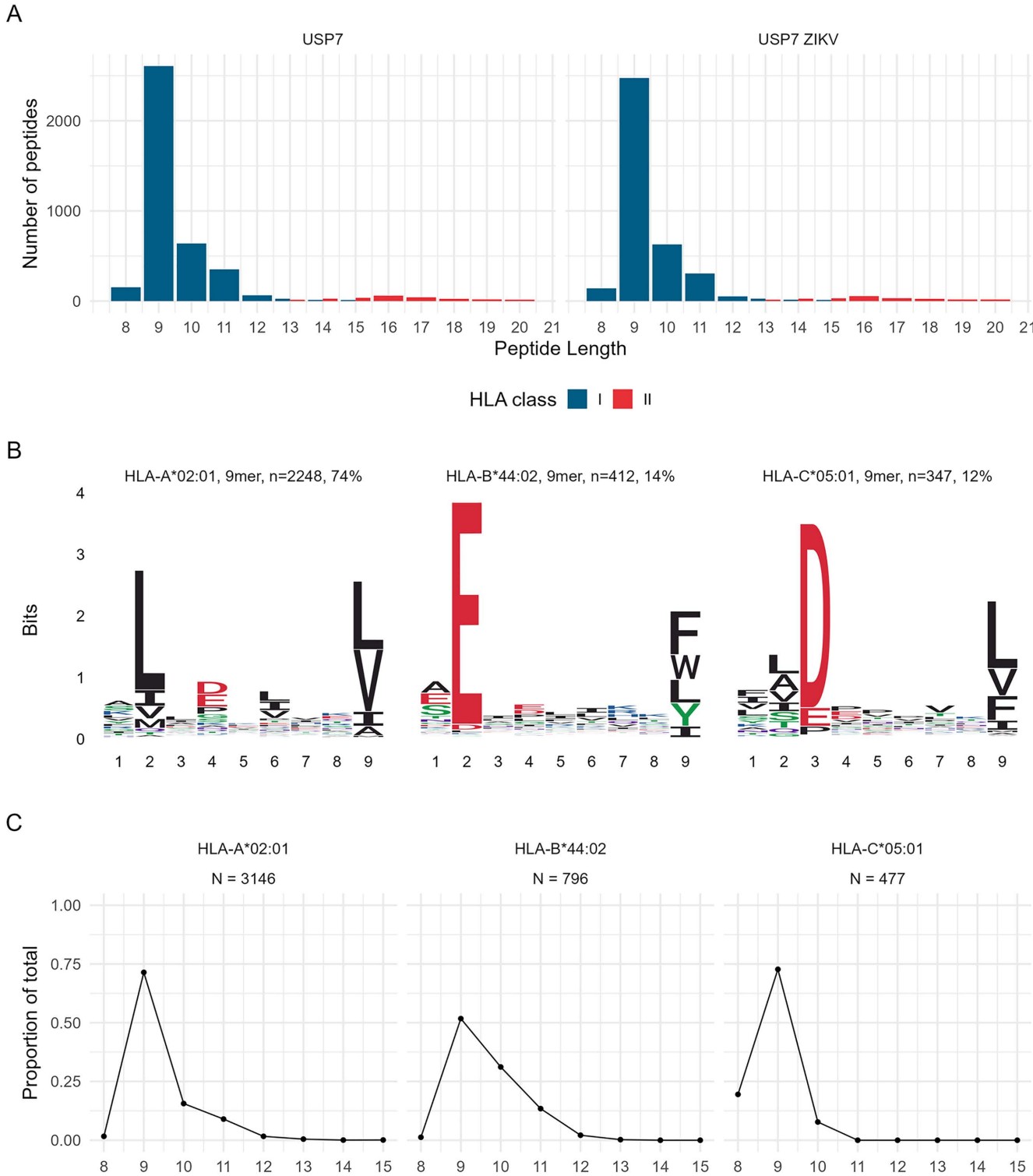

**Fig 2. The immunopeptidomes of USP7-ATRT brain tumour cells. (A)** Length distribution of HLA-I and II immunopeptides presented by Mock or ZIKV-infected USP7-ATRT cells. HLA-I peptides in blue and HLA-II peptides in red. **(B)** Class I HLA allotype 9-mer binding motifs derived from

ZIKV-infected USP7-ATRT immunopeptides by unsupervised clustering using MixMHCp. **(C)** Length distributions of peptides from ZIKV-infected USP7-ATRT cells according to clustered HLA-I allotype. Abbreviations, Zika virus (ZIKV), USP7-ATRT (USP7), human leukocyte antigen (HLA).

antigen-presenting cells, no HLA-II-presented ZIKV peptides were observed. To examine which HLA-I allotypes the observed ZIKV peptides were likely presented by, and to place these observations in the context of the host cell 9,10 and 11-mer peptides, we predicted ZIKV peptide binding affinities using NetMHC (Fig 3A, Table 4). HLA-A*02:01, HLA-B*44:02 and HLA-C*05:01 were predicted to bind and present twelve, six and one ZIKV peptides, respectively (Table 4). For HLA-A*02:01, all peptides are strong binders, and at least one of the 9, 10 and 11-mers are within the top three predicted binding peptides (Fig 3A). For HLA-B*44:02, the ZIKV NS5 9-mer is the top predicted binding peptide, all four ZIKV 10-mers are within the top seven predicted binding peptides, and the ZIKV NS3 11-mer is a low-affinity binder (Fig 3A). For HLA-C*05:01, the ZIKV NS3 9-mer is a low-affinity binder and was not plotted (Table 4). This data indicates that 17 of the identified ZIKV peptides are high-affinity binding peptides commonly within the top predicted binding peptides for each HLA allotype. Interestingly, we observe most ZIKV immunopeptides to be derived from the ZIKV NS3 followed by the NS5 RNA-dependent RNA polymerase (Table 3). To investigate the potential reason for this, we plotted the protein abundances of the ten ZIKV proteins and observed NS3 as the most abundant ZIKV protein, followed by NS5 (Fig 3B). A clear trend can be observed when protein abundances are considered alongside the ZIKV peptide number (Fig 3B, Table 3). ZIKV NS3 and NS5 are the most abundant proteins and yield the most immunopeptides, indicating that HLA-I presentation may correspond to protein abundance within its peptide processing pathway. This is further supported by the absence of immunopeptides derived from ZIKV Envelope and Membrane proteins, as HLA-I peptides are predominately derived from intracellular cytosolic proteins [30]. To conclude, HLA-A*02:01 and HLA-B*44:02 high-affinity ZIKV peptide presentation is predicted to occur on the surface of USP7-ATRT cells, and peptide presentation may correspond with protein abundance in the HLA-I pathway.

## The ZIKV NS3 helicase is a rich source of immunopeptides

Mapping the 19 ZIKV peptides onto the ZIKV polyprotein identifies two peptide-rich regions in the NS3 helicase domain and one in NS5 (Fig 4). For NS3, RMLLDNIYL, YLQDGLIASL, and LQDGLIASL overlap and reside within the 17aa sequence RMLLDNIYLQDGLIASL, whilst AEMEEALRGL and EEALRGLPVRY overlap and reside within the 14aa sequence AEMEEALRGLPVRY. For NS5, QEWKPSTGW and EEVPFCSHHF are non-overlapping and reside within the 22aa sequence QEWKPSTGWDNWEEVPFCSHHF. Performing *in silico* immunogenicity modelling of the 18 identified HLA-A*02:01 and HLA-B*44:02 presented ZIKV peptides, predicts 13 as immunogenic and five as non-immunogenic (Table 4). Interestingly, all eight ZIKV NS3 helicase peptides are predicted to be immunogenic, and out of the 18 accessed ZIKV peptides, the top three predicted immunogenic peptides are all ZIKV NS3 helicase peptides

**Table 3. Number of USP7-ATRT cell-presented ZIKV immunopeptides per protein.**

| Protein | Number of immunopeptides |
|---|---|
| NS3 | 10 |
| NS5 | 4 |
| NS4B | 2 |
| Capsid | 1 |
| NS2A | 1 |
| NS4A | 1 |

Abbreviations, Zika virus (ZIKV), non-structural (NS).

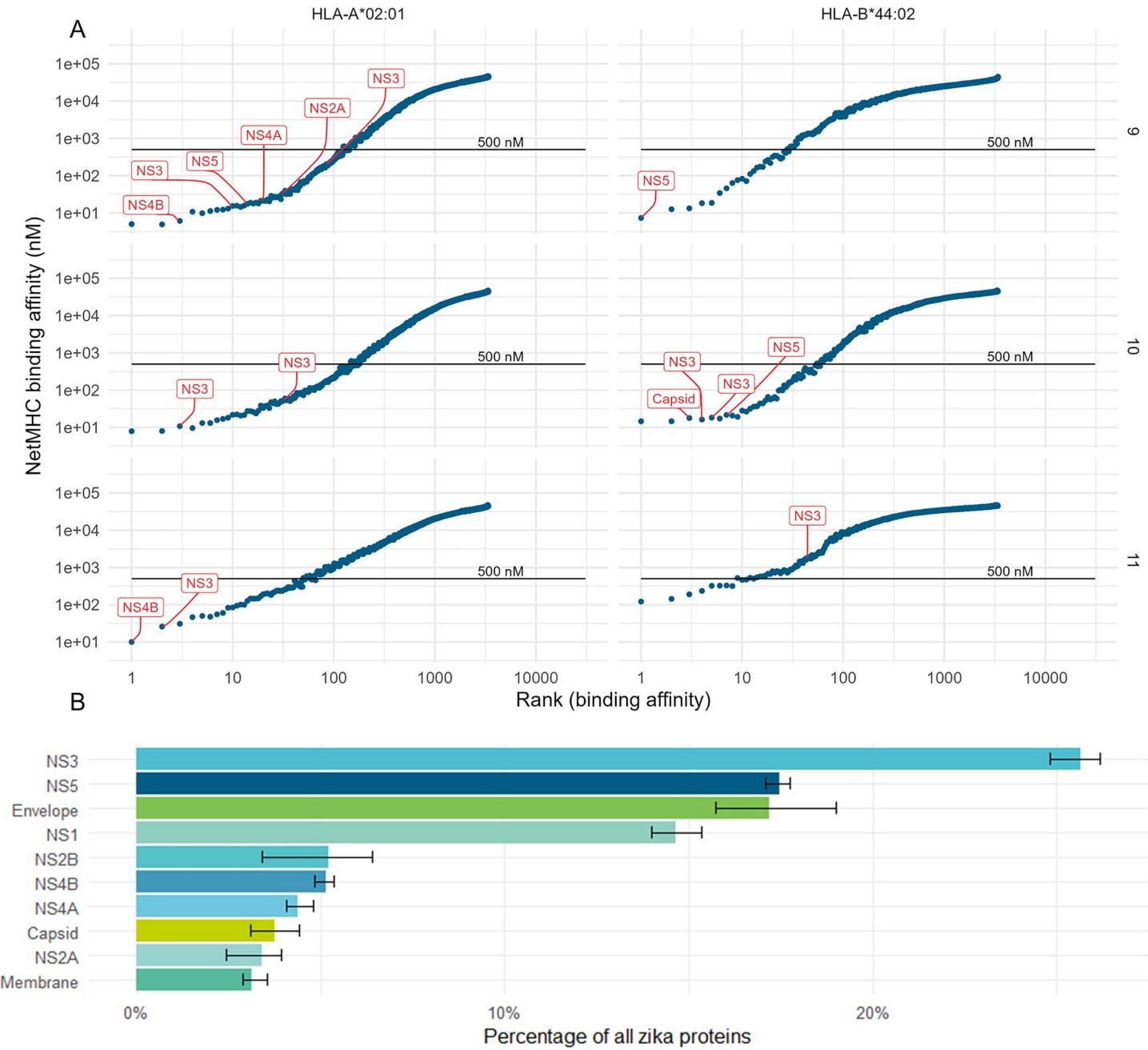

**Fig 3. ZIKV immunopeptides binding affinity and viral protein abundance. (A)** NetMHC binding predictions of all 9,10 and 11-mer ZIKV peptides (rows) to the USP7-ATRT HLA-I allotypes (columns). HLA binding affinity (y-axis) is plotted against peptide rank (x-axis). Observed peptides and their source protein are indicated in red, with the black line indicating the 500 nM threshold below which a peptide is considered a strong binder. **(B)** Barplot of the proportion of ZIKV proteins observed in the ZIKV-infected USP7-ATRT cell global proteome. Abbreviations, Zika virus (ZIKV), human leukocyte antigen (HLA), major histocompatibility complex (MHC), non-structural protein (NS).

**Table 4. USP7-ATRT cell-presented ZIKV immunopeptides and their predicted properties.**

| ZIKV Peptide | Protein | Length | *Binding affinity (nM) | **Immunogenicity | Predicted HLA |
|---|---|---|---|---|---|
| KEAMEIIKKF | Capsid | 10 | 17.9 | −0.006 | HLA-B*44:02 |
| RLVDPINVV | NS2A | 9 | 32.5 | 0.169 | HLA-A*02:01 |
| ALWDVPAPKEV | NS3 | 11 | 26 | 0.210 | HLA-A*02:01 |
| YLQDGLIASL | NS3 | 10 | 10.8 | 0.165 | HLA-A*02:01 |
| RMLLDNIYL | NS3 | 9 | 15.3 | 0.111 | HLA-A*02:01 |
| VLPEIVREA | NS3 | 9 | 296.3 | 0.368 | HLA-A*02:01 |
| LQDGLIASL | NS3 | 9 | 201 | 0.092 | HLA-A*02:01 |
| RLLGSTQVGV | NS3 | 10 | 53.3 | −0.167 | HLA-A*02:01 |
| AEMEEALRGL | NS3 | 10 | 16.3 | 0.270 | HLA-B*44:02 |
| TEVEVPERAW | NS3 | 10 | 18.3 | 0.268 | HLA-B*44:02 |
| EEALRGLPVRY | NS3 | 11 | 1796 | 0.081 | HLA-B*44:02 |
| ILDGERVIL | NS3 | 9 | 6064.7 | NA | HLA-C*05:01 |
| NQMAIIIMV | NS4A | 9 | 21.2 | 0.247 | HLA-A*02:01 |
| YLIPGLQAA | NS4B | 9 | 6.1 | −0.020 | HLA-A*02:01 |
| YLAGASLIYTV | NS4B | 11 | 10 | 0.061 | HLA-A*02:01 |
| TMMETLERL | NS5 | 9 | 10.1 | 0.186 | HLA-A*02:01 |
| SLINGVVRL | NS5 | 9 | 17.5 | 0.174 | HLA-A*02:01 |
| EEVPFCSHHF | NS5 | 10 | 21.6 | −0.078 | HLA-B*44:02 |
| QEWKPSTGW | NS5 | 9 | 7.3 | −0.331 | HLA-B*44:02 |

*Predicted by NetMHC 4.0, a binding affinity < 500 nM is usually considered as the threshold for a strong binder.

** Positive values denote predicted immunogenic peptides, and negative values denote non-immunogenic peptides. Abbreviations, Zika virus (ZIKV), non-structural protein (NS), human leukocyte antigen (HLA).

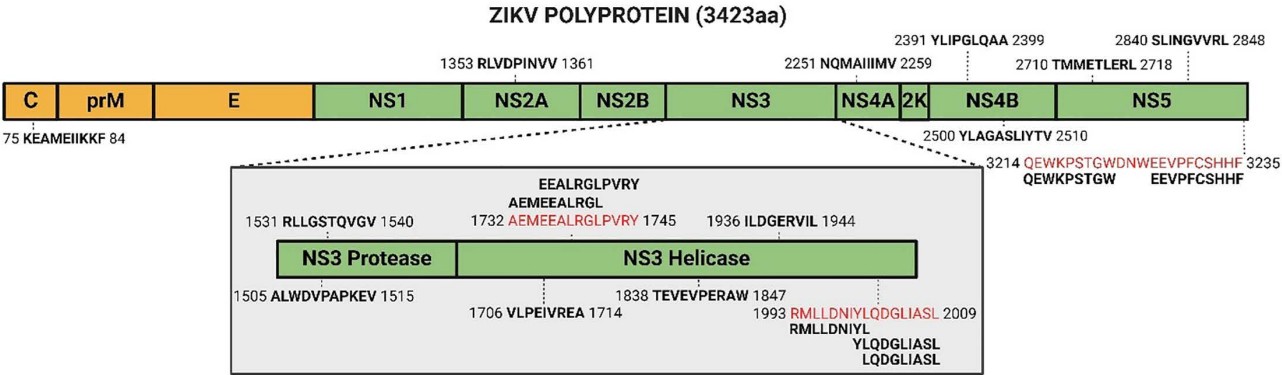

**Fig 4. ZIKV polyprotein with mapped immunopeptides.** The 19 identified ZIKV peptides are in bold, with the flanking numbers denoting the site in the polyprotein of the first and last amino acid. Peptide-rich sequences are indicated in red, with their corresponding peptides aligned above or below. ZIKV polyprotein not to scale. Figure created in BioRender. BioRender.com/r34k740BioRender.com. Abbreviations, Zika virus (ZIKV), amino acid (aa), capsid **(C)**, pre-membrane (prM), envelope **(E)**, non-structural protein (NS).

(VLPEIVREA, AEMEEALRGL and TEVEVPERAW) (Table 4). Performing peptide sequence matching, we compare our 19 ZIKV peptides to all theoretical peptides of the human proteome to assess whether they are predicted to mimic endogenous human peptides. We identify 16 ZIKV peptides to have some degree of homology with human peptides (Table 5).

**Table 5. ZIKV: human peptide homology.**

| ZIKV Peptide | Human Sequence | Protein Identifier | Gene Name | Mismatch Positions |
|---|---|---|---|---|
| ILDGERVIL | ELDGERVAL | P52429.1 | DGKE | [1, 8] |
| LQDGLIASL | LQDGLCHSL | O15021.4 | MAST4 | [6, 7] |
| RLVDPINVV | ILVDPIQVV | O75907.2 | DGAT1 | [1, 7] |
| TMMETLERL | TMMETLSRY | P21217.1 | FUT3 | [7, 9] |
| VLPEIVREA | VLPELLREA | Q92504.2 | SLC39A7 | [5, 6] |
| YLIPGLQAA | YLVPGLVAA | Q9H4B8.2 | DPEP3 | [3, 7] |
| YLQDGLIASL | YLQHGLIASA | Q5VXJ0.2 | LIPK | [4, 10] |
| AEMEEALRGL | AEEREALGGL | O15550.2 | KDM6A | [3, 4, 8] |
| EEALRGLPVRY | EEALRGLYGRV | P50053.2 | KHK | [8, 9, 11] |
| KEAMEIIKKF | KEMKESIKKF | O43615.2 | TIMM44 | [3, 4, 6] |
| NQMAIIIMV | NLMAFLIMV | Q9UBY5.1 | LPAR3 | [2, 5, 6] |
| QEWKPSTGW | QEHKPSTQN | A0A1B0GUV7.1 | TEX48 | [3, 8, 9] |
| RLLGSTQVGV | QLLLSTVVGV | P01019.3 | AGT | [1, 4, 7] |
| RMLLDNIYL | DDLLDNITL | O43524.1 | FOXO3 | [1, 2, 8] |
| SLINGVVRL | SLINFRVLL | O60287.4 | URB1 | [5, 6, 8] |
| TEVEVPERAW | TEAEVLERAN | P49189.3 | ALDH9A1 | [3, 6, 10] |
| ALWDVPAPKEV | – | – | – | – |
| EEVPFCSHHF | – | – | – | – |
| YLAGASLIYTV | – | – | – | – |

Abbreviations, Zika virus (ZIKV).

Notably, every comparison has a mismatch of at least two, and no more than seven matched amino acids are in continuous order. As eight is the minimum peptide length for binding to the HLA-I groove, none of the identified ZIKV peptides are predicted to mimic endogenous human peptides, thus supporting the predicted immunogenicity of these viral peptides. To conclude, ZIKV NS3 helicase is predicted to be a rich source of immunogenic peptides, and the identified viral peptides do not mimic theoretical peptides of the human proteome.

## Discussion

Downregulating HLA-I presentation is a mechanism which cancer cells frequently utilise to help produce an immunosuppressive TIME [31]. Here, we investigate and show that ZIKV infection enriches the HLA-I pathway in both paediatric and adult brain tumour cells. Since brain tumour cells are non-professional antigen-presenting cells, enrichment of HLA-I instead of HLA-II was expected. Supporting our observation of HLA-I pathway enrichment in brain tumour cells, HLA-A and HLA-B protein expression is significantly upregulated in ZIKV-infected U251 glioblastoma cells, where they act as a viral dependency factor and a regulator of cell viability in response to ZIKV infection, respectively [32]. We propose that the enrichment of the HLA-I pathway following ZIKV infection of brain tumour cells may contribute TIME remodelling to make these commonly immunosuppressed tumours immunogenic.

Here, we identify 19 HLA-I ZIKV peptides presented on the surface of USP7-ATRT brain tumour cells. Our work indicates the relevance of the HLA-I pathway and protein abundance to ZIKV peptide presentation and suggests a relationship as the most abundant viral proteins (NS3 and NS5) present with the greatest number of immunopeptides. The paediatric brain tumour cell line USP7-ATRT has highly advantageous traits for its use in our study because it possesses stem-like characteristics, it's highly susceptible to ZIKV infection, and it expresses the common HLA-A allotype HLA-A*02:01 [19,25]. Of the 19 peptides observed here, twelve peptides are novel, and seven have been previously observed

elsewhere. Six of our HLA-A*02:01 peptides (RMLLDNIYL, YLQDGLIASL, ALWDVPAPKEV, YLIPGLQAA, SLINGVVRL and TMMETLERL) are presented on ZIKV-infected immortalised Priess B cells that are homozygous for HLA-A*02:01 [33]. This supports our observations and indicates these six peptides as bona fide HLA-A*02:01 presented ZIKV epitopes. The peptides YLQDGLIASL, SLINGVVRL and AEMEEALRGL are recorded on the Immune Epitope Database (IEDB) under epitope IDs 2243385, 1311496 and 182464, respectively.

Our in silico immunogenicity modelling predicts 13 of the 18 identified HLA-A*02:01 and HLA-B*44:02 presented ZIKV peptides as immunogenic. A limitation of our work is that we have not validated these peptides in vitro for their ability to stimulate cytotoxic T cell responses via ELISpot assay. However, extensive literature mining revealed that four of the 13 peptides have been validated to be immunogenic by others [33–35]. NS3 YLQDGLIASL, NS4B YLIPGLQAA and NS5 SLINGVVRL stimulate memory T cell recall response in 57%, 14% and 57% of peripheral blood mononuclear cells (PBMCs) from human patients previously infected with ZIKV [33,34]. Additionally, SLINGVVRL was one of the top dominant ZIKV epitopes in an immunocompetent HLA-A2 transgenic mouse model, capable of stimulating CD8 + T cells to produce IFNγ and TNFα [35]. These observations validate our LC-MS/MS model as a viable approach to identify immunogenic HLA-I-presented ZIKV epitopes.

A limitation of our work is that we use only one cell model, which restricts the generalizability of our findings, and that USP7-ATRT cells are not optimal for considering HLA-II-presented ZIKV peptides, as they are non-professional antigen-presenting cells. There is great interest in investigating HLA-II-presented ZIKV peptides, as the infiltration and induction of CD4 + T cells contribute to ZIKV OV therapy efficacy [21,22]. This would require feeding of infected cells to professional antigen-presenting cells, such as DCs, from which immunopeptidomes could then be captured as per [26]. Microglia are the primary immune cells of the CNS, can be infected by ZIKV, and alongside DCs, possess antigen-presenting capabilities [36]. In immunocompetent mice GL261 glioma models, ZIKV administration led to increased infiltration of MHC-II-expressing microglia in the primary tumour and in long-term survivors following glioma rechallenge [22,23]. Thus, how microglia antigen presentation contributes to, and how microglia interact with CD4 + T cells during, the oncolytic response of ZIKV requires further investigation. We also note that brain tumours frequently exhibit low or reduced interferon signalling and reduced MHC class I expression, which both contribute to immune evasion. The proteomics study presented here shows that ZIKV infection can in fact increase MHC Class I expression, potentially through modulation of interferon signalling. As shown from our immuno-peptidomics study, viral replication and the associated responses can also result in the presentation of novel peptides potentially resulting in further immune activation.

It is important to understand if our ZIKV peptides have autoimmune implications by mimicking human peptides or if they may cross-react to stimulate memory T cells from previously encountered viral epitopes. Here, none of our ZIKV peptides are predicted to mimic human peptides, indicating minimal risk of autoreactive T cell activity and autoimmunity [37]. A limitation of this prediction is that the alignment is against theoretical peptides of the human proteome and, therefore, may not encompass all human peptides that may arise in vivo. The HLA-I peptides ZIKV NS3 AEMEEALRGL and NS5 SLINGVVRL are homologous to a Dengue Virus (DENV) epitope and can stimulate memory cytotoxic T cells in Japanese Encephalitis Virus (JEV)-vaccinated HLA-A2 transgenic mice, respectively [35]. It is currently unknown if and how host humoral and cellular immunity to previous viral infection or immunisation may enhance or hinder ZIKV OV efficacy. It is interesting to speculate that brain tumour cell HLA-I presentation of a ZIKV epitope may enhance OV efficacy by co-opting a patient's memory cytotoxic T cell immunity from a previous viral encounter against the tumour, but this requires investigation.

There are a multitude of complex interactions between an infected tumour and the immune system. Whilst both CD8+ and CD4 + T cells assist in glioblastoma tumour clearance, myeloid cells protect glioblastoma tumour cells from ZIKV infection through the secretion of type 1 interferons [38]. NK cells can present with dichotomous functions in response to OVs due to their contrasting antitumor and antiviral functions [39]. Thus, it cannot be assumed that all ZIKV-stimulated immune cells will promote the therapeutic properties of ZIKV, and we must elucidate the roles and interplay between these

cell types. Knowledge of the mechanisms of immune activation will factor into how ZIKV can be employed as an adjuvant for current immunotherapies. Thus far, ZIKV has proved an effective adjuvant to both immune checkpoint blockade (PD-1 and PD-L1) and vaccine-based immunotherapy to combat glioma and improve survival in mouse models [21–23]. Our identification of novel HLA-I-presented ZIKV epitopes sheds light on cytotoxic CD8 + T cell activation to contribute to the growing knowledge of how ZIKV can be employed as an immunotherapy, and may assist in the development of novel epitope-specific immunotherapies against brain tumours.

Safety concerns must be addressed for ZIKV OV therapy to progress to clinical trials, including (i) its potential for neurotoxic off-target effects and (ii) the concern of administering a replication-competent OV to immunosuppressed cancer patients. Several studies report preferential tropism of ZIKV for tumour cells over normal neural cells, and this is exemplified in canines bearing natural brain tumours where tumour tropism, but not neurotoxicity, was observed [19,20,40,41]. Recently, members of our research team genetically engineered ZIKV to express a miRNA response element that sensitises the virus to non-tumour miRNA, thereby negating ZIKV neurotoxicity by reducing viral replication in normal cells [36]. The immunodeficient nature of xenograft mice partly models the immunosuppressed nature of cancer patients. Intratumoural ZIKV administration to xenograft mice bearing brain tumours is asymptomatic with limited and non-permissive viral shedding, and does not cause adverse effects or neurological damage [19,20]. However, viral shedding and the infection of normal cells and tissues in a murine model do not fully recapitulate that of the human body. Consequently, genetic engineering of ZIKV is necessary to improve its safety profile, and current approaches have predominantly focused on employing small nucleotide changes or repurposing live attenuated vaccines [18,23,36,42–44]. Our work here identifies immunogenic regions of ZIKV which should be taken into account during future genetic engineering approaches.

Whilst our primary focus was to investigate the immunotherapeutic potential of ZIKV OV therapy, our research here is wider reaching, and has potential implications for (i) understanding fetal NPC depletion following ZIKV infection and (ii) ZIKV epitope vaccination. CZS is primarily brought about by the depletion of fetal NPCs. ZIKV infection upregulates HLA-I processing and presentation in human NPCs at the transcriptome level and in microcephalic mouse brains at the transcriptome and proteome levels [45,46]. HLA-I-presenting cells co-stain with infiltrating murine immune cells, resulting in neuronal cell death and microcephaly [46]. Thus, cytotoxic CD8 + T cell clearance may contribute to NPC depletion. To date, the immunopeptidome of ZIKV-infected NPCs is unknown, primarily due to the technical challenge of culturing NPCs to the high quantities required for immunopeptidomics. USP7-ATRT cells are of embryonal origin, closely resemble NPCs at the global gene expression level, and are immortalised so can be cultured to such quantities [19]. As such, USP7-ATRT cells are a potential model to investigate the immunopeptidome of ZIKV-infected fetal NPCs. Interestingly, HLA-C is one of only 25 genes significantly upregulated in NPCs from CZS-affected patients compared to their unaffected dizygotic twin; possibly indicating HLA-I involvement in CZS development following congenital ZIKV infection [47]. In the context of the developing fetus, the HLA-I peptides that we identify here may contribute to cytotoxic CD8 + T cell-mediated depletion of fetal NPC and subsequent CZS, and this deeply warrants investigation.

ZIKV exists as a single serotype and all strains could prove susceptible to a single vaccine [48]. Despite this, a vaccine is yet to be approved so ZIKV still poses a pregnancy risk and potential for re-emergence. Here, we identify ZIKV NS3 as a rich source of immunopeptides, producing over half of our observed epitopes. ZIKV NS3 peptides can promote NK cells and PBMC immune cell activity [33,49,50]. Additionally, ZIKV NS3 is the main antigenic T cell target, plays important roles during anti-ZIKV immunity, and a ZIKV NS3-based vaccine can stimulate the production of polyfunctional CD8 + T cells [51]. Multiple *in silico* studies employ computational approaches to propose novel ZIKV epitope vaccines, one of which predicted the NS3 helicase sequence WLEARMLLDNIYLQDGLIASLYR as the richest ZIKV polyprotein epitope source [52–55]. Our top identified region (RMLLDNIYLQDGLIASL) resides within this proposed vaccine. Additionally, we show all three ZIKV peptides within this region to be presented alongside five more HLA-I peptides within the NS3 helicase domain. Thus, we confirm the ZIKV NS3 helicase as a rich epitope source and a promising vaccine candidate.

To summarise, immunopeptidomics is a potent and powerful tool to investigate viral peptide presentation to further our understanding of the immune responses orchestrated following ZIKV infection. Our results have possible future implications for the development of ZIKV OV therapy, epitope-specific immunotherapies or ZIKV epitope vaccines.

## Materials and methods

### Cell culture and ZIKV infection

Paediatric USP7-ATRT cells were cultured as previously described [19,56]. Brazilian ZIKV KU365771 stocks were established by the Instituto Butantan (University of São Paulo, Brazil) in Vero cells and titrated by plaque-forming units (PFU) assay. For all infection experiments, USP7-ATRT cells were infected with ZIKV for 60 minutes prior to replacement with complete media. Twenty-four hours post-infection (hpi) cells were collected, washed with PBS and stored as pellets at −80°C. The infection experiments were performed in triplicate for the global proteome and once for the immunopeptidome. All controls were non-infected Mock samples.

### Transcriptomics

**HLA typing.**  High-quality total RNA was purified using the Monarch Total RNA Miniprep Kit (NEB #T2010S) as per the kit protocol from wildtype USP7-ATRT cells and sent to Novogene (UK) Company Limited for mRNA sequencing using the Illumina NovaSeq 6000 system (≥20 million 150 bp paired-end reads per sample). HLA typing was then performed using HISAT2 v2.2.0 with HISAT-genotype v1.3.0 in the default settings [57].

**Adult glioma RNA-sequencing data mining.**  RNA-Seq raw abundances of ZIKV-mCherry positive and ZIKV-mCherry negative primary cells isolated from eight GBM patients (IDs 42, 43, 45, 46, 50, 51, 54 and 57) at 72 hpi were downloaded from GEO (GSE178621) [38]. The data was normalised using DESeq2, and GSEA was performed comparing the ZIKV-mCherry positive versus negative samples, as per the ZIKV-infected USP7-ATRT global proteome analysis [58,59]. The genes which contribute to the leading edge of the GSEA analysis have been listed in S1 Table.

### Proteomics

**Global proteome sample preparation.**  50μg protein from Mock and 24-hour infected USP7-ATRT cells (MOI 2) was mixed with 600 μL methanol and 150 μL chloroform for protein extraction. 450 μL water was added to the sample, briefly vortexed and centrifuged at 14,000g for 5 min at room temperature (RT). The upper aqueous was removed and replaced with 450 μL of methanol, and the sample was then briefly vortexed and centrifuged again to pellet the proteins. The protein pellet was air-dried briefly before resuspension in 100 μL 6M urea/50 mM Tris-HCl pH 8.0. The protein was reduced by 5 mM dithiothreitol for 30 min at 37°C and alkylated by 15 mM iodoacetamide for 30 min at RT in the dark. Protein was digested with 2 μg trypsin/LysC mix (Promega) for 4h at 37°C. 750 μL 50mM Tris-HCl pH 8.0 was added, and the sample was incubated overnight at 37°C. The addition of 4 μL TFA terminated the digestion. The resultant peptide mixture was purified using HLB prime reverse phase μ-elution plates (Waters) by elution in 50 μL 70% acetonitrile according to the manufacturers' instructions, and then lyophilised.

**LC-MS/MS analysis of global proteome.**  Tryptic peptides were reconstituted in 0.1% formic acid and applied to an Orbitrap Fusion Tribrid Mass Spectrometer with a nano-electrospray ion source. Peptides were eluted with a linear gradient of 3−8% buffer B (Acetonitrile and 0.1% formic acid) at a flow rate of 300nL/min over 5 minutes and then from 8−30% over a further 192 minutes. Full scans were acquired in the Orbitrap analyser using the Top Speed data dependent mode, preforming a MS scan every 3 second cycle, followed by higher energy collision-induced dissociation (HCD) MS/MS scans. MS spectra were acquired at resolution of 120,000 at 300−1,500m/z, RF lens 60% and an automatic gain control (AGC) ion target value of 4.0e5 for a maximum of 100ms and an exclusion duration of 40s. MS/MS data were collected in the Ion trap using a fixed collision energy of 32% with a first mass of 110 and AGC ion target of 5.0e3 for a maximum of 100ms.

**Data analysis for global proteome.** Raw global proteome mass spec files were analysed using Peaks Studio 10.0 build 20190129 with spectra searched against the same database as used for immunopeptidomics. The false discovery rate (FDR) was estimated with decoy-fusion database searches and were filtered to 1% FDR. Relative protein quantification was performed using Peaks quantification module and normalized between samples using a histone ruler [60]. Downstream analysis and visualizations were mainly performed in R using associated packages [61–64]. GSEA analysis was performed on Log2(LFQ intensities) of Mock and 24-hour infected USP7-ATRT cells (N = 3) using a gene set database of Reactome innate and adaptive immune system pathways, a nominal p-value of 0.05 adjusted for multiple hypotheses testing (FDR ≤ 0.25) [65]. The genes which contribute to the leading edge of the GSEA analysis have been listed in S1 Table. The top 50 and bottom 50 GSEA-ranked proteins were plotted on a heatmap using GraphPad PRISM (10.0.3), with the ZIKV polyprotein manually ranked and incorporated. We also performed enrichment analysis using the Enrichr tool using significantly differentially-expressed sets of proteins (T-test; p < 0.05) [66].

## Immunopeptidome analysis

**Purification of HLA-I immunopeptides.** Protein-A sepharose beads (Repligen, Waltham, Mass. USA) were covalently conjugated to 10 mg/mL W6/32 (pan-anti-HLA-I) or 5 mg/mL HB145 (pan-anti-HLA-II) monoclonal antibodies (SAL Scientific, Hampshire, UK) using DMP as previously described [67]. Frozen pellets of 1x10$^8$ Mock and ZIKV-infected USP7-ATRT cells (MOI 1) were re-suspended in 5mL of lysis buffer (0.02M Tris, 0.5% (w/v) IGEPAL, 0.25% (w/v) sodium deoxycholate, 0.15mM NaCl, 1mM EDTA, 0.2mM iodoacetamide supplemented with EDTA-free protease inhibitor mix), and rotated on ice for 30 min to solubilise. Homogenates were clarified for 10 min at 2,000g, 4°C and then for a further 60 min at 13,500g, 4°C. 2 mg of anti-HLA-I conjugated beads were added to the clarified supernatants and incubated with constant agitation for 2h at 4°C. The captured HLA-I/β$_2$microglobulin/immunopeptide complex on the beads was washed sequentially with 10 column volumes of low (isotonic, 0.15M NaCl) and high (hypertonic, 0.4M NaCl) TBS washes prior to elution in 10% acetic acid and dried under vacuum. Column eluates were diluted with 0.5 volumes of 0.1% TFA and then applied to HLB-prime reverse phase columns (Waters, 30 mg sorbent/column). The columns were rinsed with 10 column volumes of 0.1% TFA and then the peptides were eluted with 12 sequential step-wise increases in acetonitrile from 2.5–30%. Alternate eluates were pooled and dried using a centrifugal evaporator and re-suspended in 0.1% formic acid.

**LC-MS/MS analysis of HLA-I peptides.** HLA peptides were separated by an Ultimate 3000 RSLC nano system (Thermo Scientific) using a PepMap C18 EASY-Spray LC column, 2µm particle size, 75µm x 50 cm column (Thermo Scientific) in buffer A (0.1% formic acid) and coupled on-line to an Orbitrap Fusion Tribrid Mass Spectrometer (Thermo Fisher Scientific, UK) with a nano-electrospray ion source. Peptides were eluted with a linear gradient of 3%−30% buffer B (Acetonitrile and 0.1% formic acid) at a flow rate of 300nL/min over 110 minutes. Full scans were acquired in the Orbitrap analyser using the Top Speed data dependent mode, performing a MS scan every 3 second cycle, followed by higher energy collision-induced dissociation (HCD) MS/MS scans. MS spectra were acquired at resolution of 120,000 at 300 m/z, RF lens 60% and an AGC ion target value of 4.0e5 for a maximum of 100ms. MS/MS resolution was 30,000 at 100m/z. Higher-energy collisional dissociation (HCD) fragmentation was induced at an energy setting of 28 for peptides with a charge state of 2–4, while singly charged peptides were fragmented at an energy setting of 32 at lower priority. Fragments were analysed in the Orbitrap at 30,000 resolution. Fragmented m/z values were dynamically excluded for 30 seconds.

**Data analysis for immunopeptidome.** Raw spectrum files were analysed using Peaks Studio 10.0 build 20190129, with the data processed to generate reduced charge state and deisotoped precursor and associated product ion peak lists which were searched against a Uniprot database (20,350 entries, 2020-04) appended with the full sequences for ZIKV strain (Brazil KU365779.1; 2015): 10 entries. A contaminants list (245 entries) in unspecific digest mode was applied [68]. Parent mass error tolerance was set a 5 ppm and fragment mass error tolerance at 0.03 Da. Variable modifications were set for N-term Acetylation (42.01 Da), Methionine oxidation (15.99 Da) and carboxyamidomethylation (57.02 Da) of cysteine. A maximum of three variable modifications per peptide were set. The FDR was estimated with decoy-fusion

database searches and were filtered to 1% FDR. The search results were further refined using the MS-Rescue package [69]. Downstream analysis and visualizations were performed in R using associated packages [61–64]. Peptide binding motifs were identified using unsupervised clustering methods MixMHCp2.1 and MoDec, for class I and class II HLA peptides, respectively [70,71]. Peptide binding affinities predicted using NetMHC 4.0 and NetMHCIIpan 4.0 for class I and class II HLA peptides, respectively [72–74]. The IEDB T cell class I Immunogenicity (1.0) tool predicted ZIKV peptide: HLA-I complex immunogenicity, with selected settings of Peptide Length(s) 9–11-mer, MHC Allele(s) HLA-A*02:01 and HLA-B*44:02, and Allele Specific anchor positions [75]. The IEDB PEPMatch (0.9) tool performed ZIKV peptide sequence matching against the human proteome, with the result specifying the best match per peptide and a maximum mismatch of three [76]. Global population coverage of HLA allotypes was sourced using the IEDB Population Coverage epitope analysis tool [77].

The mass spectrometry proteomics data have been deposited to the ProteomeXchange Consortium via the PRIDE partner repository with the dataset identifier PXD037627 and 10.6019/PXD037627 [78].

## Supporting information

**S1 Table. GSEA analyses of proteomics and RNA-Seq data.**
(XLSX)

**S1 File. Gene-enrichment analyses of proteomics data.**
(XLSX)

## Author contributions

**Conceptualization:** Matt Sherwood, Ben Nicholas, Oswaldo K. Okamoto, Paul Skipp, Rob M. Ewing.

**Data curation:** Matt Sherwood.

**Formal analysis:** Matt Sherwood.

**Funding acquisition:** Oswaldo K. Okamoto, Paul Skipp, Rob M. Ewing.

**Investigation:** Matt Sherwood, Ben Nicholas, Alistair Bailey, Thiago Giove Mitsugi, Carolini Kaid, Oswaldo K. Okamoto, Paul Skipp.

**Methodology:** Matt Sherwood, Ben Nicholas, Alistair Bailey, Paul Skipp.

**Project administration:** Oswaldo K. Okamoto, Rob M. Ewing.

**Resources:** Matt Sherwood.

**Supervision:** Oswaldo K. Okamoto, Paul Skipp, Rob M. Ewing.

**Validation:** Matt Sherwood.

**Visualization:** Matt Sherwood.

**Writing – original draft:** Matt Sherwood, Ben Nicholas, Alistair Bailey, Paul Skipp, Rob M. Ewing.

**Writing – review & editing:** Matt Sherwood, Rob M. Ewing.

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
