## [Decision Letter · Decision Letter 0]

7 Apr 2025

PONE-D-25-09932
Identification of natural Zika virus peptides presented on the surface of paediatric brain tumour cells by HLA classI
PLOS ONE

Dear Dr. Ewing,

Thank you for submitting your manuscript to PLOS ONE. After careful consideration, we feel that it has merit but does not fully meet PLOS ONE’s publication criteria as it currently stands. Therefore, we invite you to submit a revised version of the manuscript that addresses the points raised during the review process.

We look forward to receiving your revised manuscript.

Kind regards,

José Ramos-Castañeda, M.Sc., Ph.D

Academic Editor

PLOS ONE

Additional Editor Comments (if provided):

Reviewers' comments:

Reviewer's Responses to Questions

**Comments to the Author**

1. Is the manuscript technically sound, and do the data support the conclusions?

Reviewer #1: Partly

Reviewer #2: Partly

2. Has the statistical analysis been performed appropriately and rigorously? 

Reviewer #1: No

Reviewer #2: Yes

3. Have the authors made all data underlying the findings in their manuscript fully available?

Reviewer #1: Yes

Reviewer #2: Yes

4. Is the manuscript presented in an intelligible fashion and written in standard English?

Reviewer #1: Yes

Reviewer #2: Yes

5. Review Comments to the Author

Reviewer #1: Central nervous system neoplasia is associated with poor prognosis and limited therapeutic options. The use of lytic viruses with tropism for neoplastic tissue (Oncolytic viruses) is an attractive therapeutic strategy. The authors of this manuscript, among others have shown in vitro and in highly manipulated in vivo models that Zika virus, a neurotropic flavivirus, could have potential as oncolytic virus for CNS tumors.

This paper explores the global proteome and MHC-associated immunopeptidome in the surface of Zika virus-infected human pediatric brain tumor-derived cells. The main findings of the paper are 1) Components of the MHC class I pathway are enriched in Zika-infected tumor cell line, suggesting that ZV infection overcomes tumor-induced immune suppression; 2) Of 19 ZV-derived peptides presented by MHC-I, most derive from the most abundant NS3 viral protein and likely have no auto-reactive potential.

Overall, the manuscript is clearly written, and the experimental approaches are well conducted. Proteome and public transcriptome data analysis is rather superficial and needs some clarification. The main claim, the demonstration that ZV -derived peptides are presented does not imply CD8+ T cell anti-tumoral activity limit the overall contribution of the presented findings.

My specific concerns are:

1) Regarding the identification of MHC pathway enrichment by GSEA in proteome and public transcriptome analysis, authors restricted their reference pathway database to an immunity Reactome module, potentially increasing the chance of spurious enrichment. In that case a stricter P and FDR threshold would be adequate.

2) Somewhat related to MHC enrichment, flavivirus including ZV are known to inhibit type I IFN´s production and effect. Type I IFN are known potentiators of MHC expression and antigen presentation. These interactions in the context of brain tumors and the current model should be discussed.

3) 19 ZV peptides out of 3,658 (Table 2) correspond to 0.5 % of the MHC-I-eluted peptides. Is this what is expected assuming MHC-I pathway enrichment? Without a ZV-suceptible non-tumoral cell as reference, the main finding is difficult to interpret.

4) The cell line USP7-ATRT as model: Although a CNS neoplasia, atypical teratoid/rhabdoid tumors are highly undifferentiated tumors with high histologic diversity combining rhabdoid, neuroectodermal, epithelial, and mesenchymal elements. Authors describe their model as a proxy of neuronal precursor, the normal target of ZV, however it’s not clear which are the neuronal features of the USP7-ATRT cell used as model.

5) The importance of the HLA-A*02:01 allele: Authors put a considerable effort in providing evidence that the HLA-A*02:01 allele expressed in USP7-ATRT cells is highly frequent globally, as if no other alleles could also bind ZV-derived peptides and induce CTL response. In contrast with low complexity peptide vaccines of single protein vaccines, were making sure that the potential peptides can be presented by MHC in the highly diverse human population, the use of a live virus is a different scenario: There are at least several antigenic viral proteins, which are likely to be presented by different HLA genes and allele combinations in the global population.

Thus, the argument of HLA-A*02:01 allele being the most frequent in the global population is not necessary, as it is also unnecessary to estimate the binding affinities of the identified peptides with the HLA-A*02:01 allele. In fact, it would be more informative to estimate binding affinities of the identified peptides with other globally common HLA alleles.

Moreover, authors should provide the exact source of the HLA-A*02:01 allele frequency reported in table 1. Searching the Allele Frequency Net Database http://www.allelefrequencies.net/, HLA-A*02:01 allele is more frequent in European population (but not 40%, as stated), but not so in other major populations. Also, reported frequencies are subjected to strong population sampling biases, being the European population the most sampled.

Minor concerns:

1) Line 105. “via vertical transmission (12)”. I suggest changing to “transplacental transmission”, since vertical transmission implies the transmission of genetic information through the germline.

2) Line 293. “Downregulating HLA-I presentation is a method…”. I suggest changing to “mechanism”.

3) Microglial cells have antigen-presenting functions via class II and are targets of ZV. Should be discussed in the context of oncolytic virus therapy.

Reviewer #2: In this manuscript, the authors identify and characterize Zika virus-derived peptides, mostly from the NS3 protein, naturally presented by HLA class I molecules on a brain tumor cell line USP7-ATR, using immunopeptidomics, highlighting the enrichment of the antigen presentation pathway and proposing these peptides as novel, non-self targets for cytotoxic T cell responses in the context of oncolytic virotherapy and cancer immunotherapy.

This is a very good paper, well written and pleasant to read. I don’t have major concerns or specific questions, but I would like to raise a few points for consideration. First, no T cell assays (e.g., ELISpot or cytotoxicity assays) are included to confirm the immunogenicity of the identified peptides. Although the authors reference previous studies, would it be feasible to include such data to directly demonstrate T cell activation in response to the novel peptides identified here? Additionally, immunopeptidomics was performed only once (line 417), which limits the reproducibility of the peptide identification, and the manuscript does not acknowledge this limitation. The findings are also based on a single cell line (USP7-ATRT), which, while appropriately justified as a relevant model, restricts the generalizability of the results. Could the study be extended to include additional cell lines? Another limitation is the lack of experimental validation for affinity and stability of HLA-peptide binding, which is based solely on computational prediction.

In conclusion, this is an interesting and well-written manuscript. Including additional experimental validation, as suggested above, would further strengthen the message and impact of the study.

6. PLOS authors have the option to publish the peer review history of their article (what does this mean?). If published, this will include your full peer review and any attached files.

Reviewer #1: No

Reviewer #2: No

---

## [Author Response · Author response to Decision Letter 1]

16 Sep 2025

See attached response to reviewers file.

---

## [Decision Letter · Decision Letter 1]

16 Oct 2025

Identification of natural Zika virus peptides presented on the surface of paediatric brain tumour cells by HLA classI

PONE-D-25-09932R1

Dear Dr. Ewing,

We’re pleased to inform you that your manuscript has been judged scientifically suitable for publication and will be formally accepted for publication once it meets all outstanding technical requirements.

Kind regards,

José Ramos-Castañeda, M.Sc., Ph.D

Academic Editor

PLOS ONE

Additional Editor Comments (optional):

Reviewers' comments:

Reviewer's Responses to Questions

**Comments to the Author**

1. If the authors have adequately addressed your comments raised in a previous round of review and you feel that this manuscript is now acceptable for publication, you may indicate that here to bypass the “Comments to the Author” section, enter your conflict of interest statement in the “Confidential to Editor” section, and submit your "Accept" recommendation.

Reviewer #1: All comments have been addressed

2. Is the manuscript technically sound, and do the data support the conclusions?

Reviewer #1: Yes

3. Has the statistical analysis been performed appropriately and rigorously? 

Reviewer #1: Yes

4. Have the authors made all data underlying the findings in their manuscript fully available?

Reviewer #1: Yes

5. Is the manuscript presented in an intelligible fashion and written in standard English?

Reviewer #1: Yes

6. Review Comments to the Author

Reviewer #1: All issues raised in my first review have been convincingly addressed. Therefore I consider that the manuscript is suitable for publication in PLoS One.

7. PLOS authors have the option to publish the peer review history of their article (what does this mean?). If published, this will include your full peer review and any attached files.

Reviewer #1: No

---

## [Editor Report · Acceptance letter]

PONE-D-25-09932R1

PLOS ONE

Dear Dr. Ewing,

I'm pleased to inform you that your manuscript has been deemed suitable for publication in PLOS ONE. Congratulations! Your manuscript is now being handed over to our production team.

Kind regards,

on behalf of

Dr. José Ramos-Castañeda

Academic Editor

PLOS ONE